# Body Fat Is a Predictor of Physical Fitness in Obese Adolescent Handball Athletes

**DOI:** 10.3390/ijerph17228428

**Published:** 2020-11-14

**Authors:** Souhail Hermassi, Nicola Luigi Bragazzi, Lina Majed

**Affiliations:** 1Sport Science Program, College of Arts and Sciences, Qatar University, Doha 2713, Qatar; lina.majed@qu.edu.qa; 2Postgraduate School of Public Health, Department of Health Sciences (DISSAL), 16132 Genoa, Italy; robertobragazzi@gmail.com or; 3Laboratory for Industrial and Applied Mathematics (LIAM), Department of Mathematics and Statistics, York University, Toronto, ON M3J 1P3, Canada

**Keywords:** physical performance, handball, anthropometrics, body fat, body mass index, obesity

## Abstract

We examined the relationships between body fat (BF) and field measures of physical fitness in adolescent handball players. Twenty nine players (age: 16.6 ± 1.72 years; body mass: 79.8 ± 17.0 kg; height: 1.70 ± 0.12 m; body fat: 27.7 ± 8.67%) from Qatar handball first league performed a series of anthropometric and fitness tests related to their performance in sprinting (i.e., 15 m and 30 m sprint), jumping (i.e., countermovement and squat jumps), throwing (i.e., 3 kg medicine ball seated front throw) and running (i.e., agility T-Half and Yo-Yo intermittent recovery level 1 tests). Significant differences between obese and non-obese groups, classified based on age-stratified %BF norms, were found, with the largest difference being attributed to aerobic performance on the Yo-Yo test. Results indicated no significant relationships between anthropometric variables and sprinting or jumping abilities. %BF predicted a significant 8–15% portion in running performances of agility and aerobic capacity, while the latter were mainly explained by body height and mass. %BF was the only and strongest predictor of throwing performance, being an important determinant of performance in handball. Optimizing %BF should be considered as a training and nutrition goal in order to improve sport performance.

## 1. Introduction

Handball is an Olympic sport played worldwide and at a highly professional level in many countries. Handball is an indoor team sport relying on high-intensity intermittent activities with increased demands for muscular strength, explosive strength, speed, agility, reactive agility, power, flexibility and muscular endurance [1,2]. With regards to anthropometric characteristics, team handball players were characterized by values of body mass, height, body fat percentage (%BF) and body mass index (BMI) similar to those of team sport players (e.g., Rugby and American football) [3,4,5,6,7]. Recent studies have suggested that anthropometric measurements might be related to physical fitness components in team sports [1,2,3,4,5]. For example, high body mass and %BF measurements were related to poor muscle power in soccer [8], basketball [9], and handball [10] players.

In addition, a sport-specific performance characteristic of team handball [11] and one of the training and nutrition concerns in handball is the optimization of body composition. An assessment method to monitor body mass status was body mass index (BMI), which has been used to categorize humans as underweight normal weight, overweight, and obese for health purposes. BMI was included in several physical fitness batteries administered to handball players [2,12,13,14,15]. Although BMI was found to significantly influence individual fitness in youth [16], this indicator is less often utilized in athletic populations. Given that the body mass status might vary by age (i.e., lower prevalence in the older age groups) in team sports, such as handball [17], it becomes important to consider %BF, rather than BMI, when investigating the relation to fitness. Previous reports have examined the association between anthropometric measures and physical fitness in children and adolescents. Generally, a negative correlation between %BF and performance on several fitness tests (e.g., fist ball throw, standing long jump, 30 m dash and 1200 m running) was indicated in 9–13 year-old boys [15] and specifically in cardiorespiratory fitness in youths above the age of 8 years [18,19]. However, most studies investigating this relationship have focused on non-athletic populations with an emphasis on health rather than performance characteristics.

From the literature, it is known that overweight and obese adolescents generally have a poorer performance in motor tests than their normal weight counterparts [20]. A lot of previous research focused on adolescents has proven that overweight and obesity negatively affect the level of fundamental motor skills and motor performance [21,22]. Specifically, severe deficits in fundamental movement skills, diagnosed as developmental coordination disorder, were significantly linked to a high percentage of body fat [23,24]. Although the abovementioned research enhanced our understanding on the relationship between anthropometry and physical fitness, this topic has not been studied in handball players so far. Considering the popularity of handball as the number two sport in the world, this information would have practical applications for practitioners working with handball players (e.g., coaches and fitness trainers), especially with adolescents. It should be highlighted that physical fitness components such as aerobic capacity and sprints have been shown to discriminate handball players of different competitive levels [25]. Furthermore, it is needed to clarify the associations between anthropometric characteristics (i.e., body height and mass and %BF) and physical fitness in adolescent handball players, reflecting performance parameters.

Consequently, the results in this direction seem the potential benefits of long-term athletic development as a pathway that could enhance the health, fitness, and performance of adolescent players [26,27]. However, the development of sporting talent is very important, highly valued, and extremely rewarding for both athletes and practitioners alike; however, it is imperative from a public health perspective that a structured, progressive, and integrated approach to youth training is viewed as a developmental pathway for players and adolescents of all ages and abilities. Likewise, this research area scarcely concerns countries from the Gulf region, and this geographical area has unique features, which means that research from other parts of the world may not be applicable to the Gulf.

It seems that an increased %BF presents different implications for running and jumping performances among different age athletes and this must be taken into account by coaches and trainers. Furthermore, there is a need to clarify the associations between factors such as %BF and fitness affecting the physical performance in team handball players. The overall overweight and obesity prevalence among male adolescent in Qatar was higher [28] than the global prevalence of obesity and overweight (18%) reported by the WHO in 2016. Due to sedentary lifestyles, the number of overweight adolescents in Qatar is increasing every day while their level of physical activity and motor skills is declining.

Therefore, the aim of this study was to investigate the relationship between anthropometric measurements (i.e., body height and mass and %BF) and field measures of physical fitness in adolescent handball players, reflecting performance parameters. The research hypothesis was that body fat would be negatively associated with performance in sprinting, jumping, throwing and running tests, even after accounting for other basic anthropometric measures like body height and mass. The contribution of body fat is expected to be higher for performances requiring weight-bearing exercises (i.e., running), reflecting aerobic capacity and agility.

## 2. Materials and Methods

### 2.1. Participants

The investigation was completed in an in-season period, from January to February 2020. Twenty-nine male adolescent handball players with a playing experience average of 5.05 ± 0.94 years, from the Qatar handball first league were recruited for the study (4 goalkeepers; 6 pivots; 12 backs; 7 wings). Six players were left handed and 23 players were right handed. Twenty-nine male adolescent handball players from the Qatar handball first league were recruited for the study. Their physical characteristics are presented in Table 1. Participants did not report any musculoskeletal injuries in the 4 weeks prior to the commencement of the study. Players trained on average 4.2 ± 0.05 h week^−1^, had an experience of at least four years and routinely competed in one match during the weekend. Training consisted mainly of tactical skill development (60% of session time) and strength and conditioning routines (40% of session time). They did not train between the pre-test and 24 h after performing the testing procedures to avoid any strenuous exercise on the day before testing, and no additional training was conducted on the testing days. Exclusion criteria included the existence of any chronic disease or orthopedic condition that might interfere with the participation in the training program, matches, or experimental tests. Prior to the start of the study, all players and/or their guardians signed a written informed consent and/or assent in accordance with the Declaration of Helsinki. The present study’s protocol was approved by the university’s institutional review board (QU-IRB 1163-EA/19).

### 2.2. Procedures and Measurements

All testing sessions were performed in a standard indoor handball court, at the same time of day (from 6:00 p.m. to 8:00 p.m.) and under similar environmental conditions (temperature: 22.5 ± 0.5 °C, relative humidity: 60 ± 5%), at least three days after a competition. Participants were advised to maintain their normal dietary habits, while refraining from drinking caffeine-containing beverages or eating for 4 h and 2 h before testing, respectively. They were also asked not to perform any vigorous physical activity for 24 h before testing. Tests were performed over a period of four days in a fixed order to elicit similar fatigue effects between players. On the first day, anthropometric measurements were followed by vertical jump tests (SJ and CMJ). The second day was devoted to sprint performance. On the third day, the agility T-Half test and medicine ball throw were evaluated, followed by the Yo-Yo Intermittent Recovery test level 1 (Yo-Yo IR1) on the fourth day.

Two weeks after the initial testing period, fitness tests from day 1 to day 3 were repeated (i.e., SJ, CMJ, sprints, T-Half agility, medicine ball throw) to allow the assessment of the test-retest reliability of measurements. Scores of the second set of tests were considered for analyses. Anthropometric assessments and the Yo-Yo IR1 test were only done once at the initial testing period for convenience reasons related to time and players’ schedules.

### 2.3. Anthropometry

Anthropometric measurements included standing height (Holtain stadiometer, Crosswell, Crymych, Pembrokeshire, United Kingdom, accuracy of 1 mm) and body mass (model TBF 105; Tanita Corporation of America, Inc, Arlington Heights, IL, USA) and were measured to the nearest 0.1 cm and 0.1 kg, respectively. Body mass index (BMI) was calculated as the ratio between body mass (kg) and body height squared (m^2^). Percent body fat (%BF) was assessed using the skinfold method with a Harpenden caliper (Baty International, Burgess Hill, Sussex, United Kingdom). Skinfold thickness was measured to the nearest 0.1 mm with a Harpenden caliper. Duplicate readings were taken at each site, and the average of the two was recorded. If the two readings differed by more than 2 mm, a third one was taken, and the closest two were averaged. The sum of the four skinfold measurements was used as an estimate of body fat according to the sex- and age- specific Durnin-Womersley equation [29] as previously reported in adolescent handball players [30]. However, the overall percentage of body fat was estimated from the biceps, triceps, subscapular, and suprailiac skinfolds, using the equation [29]:% Body fat = (4.95/(Density − 4.5)) × 100
where Density = 1.162 − 0.063 × (LOG sum of 4 skinfolds).

Specifically, cut-off values were different for participants aged 17 years and younger (i.e., 31% BF for obesity), compared to those aged 18 to 19 years (i.e., 22% BF for obesity) [31].

### 2.4. Squat (SJ) and Counter Movement Jump (CMJ) Tests

Prior to the jumping tests, the participants followed a similar general warm-up procedure that included 5 min of running, stretching of lower limbs muscles and 2 min of jumping exercises. SJ and CMJ heights were assessed using the Optojump photoelectronic cells (Optojump Next, Microgate, Italy) [32]. Jump heights were measured from the recorded contact and flight time of vertical jumps with an accuracy of 1/1000 s (1 kHz). The SJ began at a 90° knee angle; avoiding any downward movement, participants performed a vertical jump by pushing upwards with their legs. The CMJ began from an upright position, with participants making a rapid downward movement to a knee angle of approximately 90°, arms akimbo and simultaneously beginning to push-off, after being instructed to jump as fast and high as possible. Both tests were performed without an arm swing by keeping the hands fixed at the level of the pelvis and with knees and ankles extended at take-off and landing. The largest of four jumps was recorded for each test, and a 30 s recovery was given between each jump.

### 2.5. Sprint Tests

Prior to sprint testing, each participant performed a 5 min warm up, consisting of 3 min of running, change of direction activities and dynamic stretching. Participants ran 15 m from a standing position, with the front foot 0.2 m behind the starting photocell beam. Times at 15 m and 30 m were recorded by paired photocells (Racetime 2 SF, Microgate, Bolzano, Italy) that were located 1 m above the ground at the start and finish lines. Three trials were separated by 6 to 8 min of recovery, and the fastest trial was retained for further analyses.

### 2.6. Ability to Change Direction (T-Half Tests)

A 10-min warm-up including jogging, lateral displacements, dynamic stretching and jumping was done prior to the tests. T-Half tests [33] data were recorded using electronic timing sensors (photocells, Kit Racetime 2 SF, Microgate, Bolzano, Italy) set at 0.75 m above the floor, 3 m apart and facing each other at the starting line A. Testing began with the front foot placed 0.2 m behind line A. At their discretion, players sprinted forward to cone B and touched its base with their right hands. Facing forward and without crossing feet, they shuffled to the left to cone C and touched its base with their left hands. They then shuffled to the right to cone D and touched its base with their right hands, subsequently running back to the left to cone B and touching its base. Finally, they ran backwards as quickly as possible, returning to line A. Anyone who crossed one foot in front of the other, failed to touch the base of a cone, and/or failed to face forward throughout had to repeat the test. Participants repeated the test until two successful trials were done, with 3 min of rest between trials, and only the best trial was considered in the analyses.

### 2.7. Medicine Ball Overhead Throw

Prior to the medicine ball overhead throw testing, each subject performed a 5-min warm up, consisting of 3 min of running and dynamic activities. Medicine ball throws were performed using 21.5 cm diameter 3 kg rubber medicine balls (Tigar, Pirot, Serbia). All subjects began with a familiarization session. A brief description of the optimal technique was given, suggesting a release angle to achieve a maximum distance of throw [34]. The sitting player grasped the medicine ball with both hands, and on the given signal forcefully pushed the ball from the chest. The score was measured from the front of the sitting line to the place where the ball landed. Three trials were performed and separated with one minute of rest. Criteria for an acceptable test were recording of the better of two definitive trials. The distance of the throw was recorded in 0.01 m.

### 2.8. The Yo-Yo Intermittent Recovery Test Level 1 (Yo-Yo IR1)

The Yo-Yo IR1 was conducted as described by Krustrup and collaborators [35]. The reliability of the test has been established with a coefficient of variation of 3.6% and an intraclass correlation (ICC) coefficient of 0.94 [35]. A standardized warm up comprised 5 min of low-intensity running. Participants performed 20-min shuttle runs at increasing velocities until exhaustion, with 10 s intervals of active recovery (2 × 5 m of jogging) between runs. The test was terminated if participants failed twice to reach the front line in time (objective criterion) and/or felt unable to complete another shuttle at the required speed (subjective criterion). The total distance covered was considered as the test score.

### 2.9. Statistical Analyses

Prior to inferential statistical analyses, normality of all descriptive values was evaluated by visual inspection of quantile-quantile (Q-Q) plots [36] and with Shapiro–Wilk tests. Descriptive statistics were presented as mean, standard deviation (SD) and 95% confidence intervals (95% CI). Intrarater reliability was assessed for each of the fitness tests between the first and second testing sets using ICC [37]. Interpretation of ICC values was based on guidelines provided by Portney and Watkins [38], where a value above 0.75 indicates good to excellent reliability, while below 0.75 reliability is considered poor to moderate [38]. Independent *t*-tests examined the differences in all variables between the obese and non-obese groups. Homogeneity of variances was assessed using the Levene’s Test for Equality of Variances. Multiple hierarchical regressions were then used to investigate the relationship between anthropometric variables and performance parameters (i.e., scores of fitness tests). Body height and mass were entered in the first step, and %BF was entered in the second step to examine its unique contribution to each performance parameter after accounting for the variance explained by body height and mass. Prior to the regression analyses, assumptions of linearity (scatterplots), multi-collinearity between predictors, homogeneity of variance and independence of observations (Durbin-Watson test) were verified to ensure that results are representative. Statistical analyses were performed using SPSS version 25.0 for Windows (SPSS Inc., IBM, Armonk, NY, USA).

## 3. Results

### 3.1. Intrarater Reliability

All variables showed an excellent intrarater reliability (Table 2). The lowest ICC value was 0.98 (i.e., SJ, Agility T-Half, Medicine ball throw 3 kg). Only the parameter SJ displayed a very wide 95% CI with a lower limit below 0.75 (ICC = 0.45).

### 3.2. Performance Parameters by %BF Group

No significant differences were found between the non-obese and obese groups in age, body height, body mass or even BMI (*p* > 0.05, Table 1). Groups differed significantly in terms of total body fat (%) and in all skinfold measures (*p* < 0.001, Table 1). Most performance parameters were systematically higher in the non-obese group as compared to the obese group (Table 3). Compared to the non-obese group, obese participants had performances that were on average 8 to 9% lower on the 15 m sprint, the agility T-Half test and the medicine ball throw, while this difference reached about 30% for the Yo-Yo IR1 test.

### 3.3. Hierarchical Regressions

Hierarchical multiple regressions were conducted with performance parameters as dependent variables. Body height and mass were entered at the first step and %BF at the second to account for its unique contribution to performance. Results indicated that anthropometric variables were not able to predict the sprint performances (i.e., 15 m and 30 m sprint tests, Table 4). Body height and body mass (step 1) contributed significantly to explaining 36%, 41% and 64% of the variance in the SJ, agility T-Half, and Yo-Yo IR1 performances, respectively. The addition of %BF to the regression models (step 2) explained an additional 5%, 9%, 37% and 15% of the variance in scores of SJ, agility T-Half, medicine ball throw and Yo-Yo IR1 scores and those changes in R^2^ were significant (Table 4). When all three anthropometric variables were included, a significant 51% of the variance in CMJ scores was accounted for.

## 4. Discussion

The aim of this study was to examine how body fat contributes to explaining variations in fundamental field measures of physical fitness in adolescent handball players. Findings confirmed previous observations about the negative effect of high %BF on both aerobic and anaerobic fitness components, confirming thereby our first hypothesis [38]. After accounting for body height and mass, %BF was still able to explain a significant portion of specific performance parameters. Indeed, body height and mass played an important role in agility and aerobic capacity testing (i.e., running tests), while for instance medicine ball throw seemed to be mainly explained by %BF. The latter only partially supports our second hypothesis for the significant relationship between %BF and scores on running tests (i.e., weight-bearing). The present study was able to differentiate between the unique contribution of %BF to performance parameters in young handball players.

Participants were divided according to age-stratified cut-off values of %BF, either as obese or non-obese. BMI values of both groups did not differ, while significant differences existed in %BF and all skinfold measurements. This result shows that BMI is a weak indicator for identifying obesity in athletic populations [39,40].

The first finding of the present investigation was the significantly lower performance associated with high %BF on several fitness tests in adolescent handball players. When players were categorized based on age-stratified %BF norms, anaerobic performance in sprinting (i.e., 15 m sprint), throwing (i.e., medicine ball) and agility (i.e., T-Half test) tests were 8–9% lower for the obese group as compared to the non-obese one. However, obesity seemed to affect the aerobic performance to a larger extent, where a significant 30% difference was found between groups. This result is in agreement with previous studies indicating that having more body fat is associated with lower aerobic fitness in children and adolescents [41,42,43]. This relation was relatively well investigated in the general youth population mostly for health purposes [42,44], however fewer reports focused on sport populations from a performance perspective. For instance, performance of elite female athletes in a pentathlon was found to be mostly determined by body fatness as compared to components of bones and muscles [45]. Nevertheless, other findings also indicated that the negative correlation between %BF and $\dot{V}$O_2max_ was not significant in 25 female athletes of 17–22 years old [46]. In adult male soccer players and youth basketball players, a negative effect of fatness was found on selected parameters of physical fitness, including both anaerobic and aerobic components [7,8].

The aforementioned results on the groups’ comparative analyses was based on a simple dichotomic categorization relative to pre-defined %BF cut-off values found in the literature [31]. Although findings are in agreement with the literature, it remains important to take the analysis a step further and consider the broader influence of %BF, as a continuous variable, on performance characteristics. The possible differences in playing experience (e.g., range: 3–6 years) might have influenced body composition, since players tended to have a higher %BF than the other players. Furthermore, handball-playing position can influence the amount of total %BF as many authors described that players are significant different regarding body mass depending on the playing position body, weight, BMI, arm span and palm length. Fieseler al. [2] reported that backs, pivots and goalkeepers present remarkably greater body mass segments than wings, which are the shortest players with the least body weight

Stepwise regressions investigating the unique contribution of %BF to essential performance features in handball offered a better insight into the specific role of body fatness after controlling for the influence of body height and mass that plays an important role in any physical testing. Findings revealed no significant relationship between anthropometric variables and sprinting performances (i.e., 15 m and 30 m sprints). Furthermore, jumping performances (i.e., CMJ and SJ) were mostly explained by body height and mass (step 1) with no additional influence of %BF (step 2). The latter suggests that previous reports showing significant correlations between BMI and physical fitness do not capture the influence of body fatness separately, and interpretations relative to overweight and obesity based on BMI classifications would merit caution in athletic populations.

Another main result of the regression analyses is the significant association between anthropometric variables and throwing and running tests. Body height and mass were the strongest predictors of scores on the agility T-Half and Yo-Yo tests and were able to predict alone 41% and 64% of the variance, respectively. The addition of %BF in step 2 showed a significant unique contribution of 9% and 15% to the latter performances, respectively. Interestingly, when considering the performance on the medicine ball throwing test, %BF was the strongest predictor of the throwing performance (i.e., medicine ball throw) explaining alone 37% of its variance, while body height and mass did not show a significant predicting power.

An important finding was obtained from the comparison of the two groups with different %BF, which revealed that the highest body fat scored lower in most of the tests. This suggests that a threshold might exist in body fat, above which physical fitness is affected largely. Moreover, regression analyses indicated that regardless of set thresholds, an increase in %BF would mostly negatively impact performances not only on throwing, but also on running, both of which are essential components of team handball.

### Limitations of the Study

The age range of participants in the present study varied between 15 and 19 years. Although groups were created based on age-stratified %BF norms, sexual maturation was not considered. A previous study indicated that maturation status explains a small part of the variance in aerobic fitness independent of %BF [18]. Future studies should include an assessment of sexual maturation when examining the relationship between anthropometry and physical fitness in youth. Since the design of this study was cross-sectional, it is hard to investigate the causality in the studied relation. Finally, it would be advantageous for further studies to include laboratory tests of fitness relative to the studied population, rather than field tests. Finally, a closer look at the fat distribution (e.g., from several skinfold measures) and its relationship to fitness variables would add an interesting perspective and a more focused analysis to understanding performance.

## 5. Conclusions

Adolescent handball players from the Qatar handball first league presented a %BF of 27.66 (±8.67), which is considered relatively high for young athletic populations. Therefore, adolescent handball players with an obesity classification based on %BF showed lower performance in fundamental physical fitness tests. Specifically, the increase of %BF seemed to have deleterious effects on throwing and running performance after controlling for body height and mass. Optimizing and decreasing body fat should be considered as a training and nutrition goal in order to improve not only the running performance (aerobic capacity and agility), but anaerobic components such as throwing, sprinting and jumping. These findings may prove very helpful for the assessment and evaluation of talents and should help to develop and optimize trainings regimes. We assume that not only body mass status, but also the ratio between the amount of body fat and lean mass plays a crucial role in developing athletes. BMI measurements are unable to differentiate between fat and lean masses, and therefore can be considered a poor indicator in predicting components of physical performance in handball. Major attention should be paid to multi-factorial parameters to further deepen the investigation of intra- and interrelationships between physiological and anthropometric characteristics, including body fat percentage, and the performance of adolescent handball players.

### Practical Application

The findings underline the importance of implementing programs targeting weight/fat control for enhancing and optimizing sports performance. Therefore, handball clubs and other sports institutions should place the onus on developing exercise-training and conditioning programs that are specifically aimed at controlling body mass and fat. In this study, an attempt to elucidate the effects of excessive fatness on performance parameters among the young handball players is suggested to devise and implement ad-hoc exercise interventions aimed at targeting and managing body mass status. The above results can prove to be useful for coaches and trainers, to develop position-specific training concepts as they are based on tests, which reflect the specific characteristics of individual playing categories. These findings could be added to the international literature and assist in talent identification and development.

## Figures and Tables

**Table 1 ijerph-17-08428-t001:** Participants’ physical characteristics by category of percent body fat (non-obese and obese). Data are presented as means (standard deviation).

Physical Characteristics	Non-Obese	Obese	Total
	(*n* = 16)	(*n* = 13)	(*n* = 29)
Age (years)	16.44 (1.93)	16.69 (1.49)	16.55 (1.72)
Body height (cm)	169.75 (14.25)	171.15 (8.81)	170.38 (11.94)
Body mass (kg)	74.62 (13.49)	86.15 (19.10)	79.79 (16.97)
BMI (kg·^−2^)	26.12 (5.24)	29.49 (6.78)	27.63 (6.11)
Body fat (%)	21.66 (4.70)	35.04 (6.37) ***	27.66 (8.67)
Skinfold measurements			
Bicipital skinfold (mm)	10.12 (4.50)	41.69 (25.71) ***	24.28 (23.44)
Tricipital (mm)	17.12 (7.54)	49.23 (20.56) ***	31.52 (21.97)
Subscapular (mm)	21.31 (12.25)	55.38 (22.98) ***	36.59 (24.58)
Suprailiac (mm)	14.62 (6.26)	56.54 (23.02) ***	33.41 (26.42)

BMI: body mass index. *** significant difference between groups, *p* < 0.001.

**Table 2 ijerph-17-08428-t002:** Intrarater reliability of each performance parameter tested for handball players (*n* = 29). Means and standard deviation are presented for both testing sets, with the intraclass correlations (95% confidence interval).

Tests	1st Testing Set	2nd Testing Set	ICC (95% CI)
Sprint 15 m (s)	2.93 ± 0.30	2.97 ± 0.30	0.99 (0.97–1.00)
Sprint 30 m (s)	5.18 ± 0.86	5.19 ± 0.86	1.00 (0.99–1.00)
Countermovement jump (cm)	32.1 ± 4.42	31.5 ± 4.45	0.99 (0.88–1.00)
Squat jump (cm)	25.2 ± 4.31	24.3 ± 4.08	0.98 (0.45–1.00)
Agility T-Half (s)	7.73 ± 0.98	7.84 ± 0.97	0.98 (0.96–0.99)
Medicine ball throw 3 kg (m)	8.08 ± 0.76	8.04 ± 0.70	0.98 (0.97–0.99)

**Table 3 ijerph-17-08428-t003:** Comparisons of performance parameters between the non-obese and obese groups.

Tests	Non-Obese	Obese	*t*-Statistic
	(*n* = 16)	(*n* = 13)	
Sprint 15 m (s)	3.08 (0.30)	2.81 (0.24) *	*t* (27) = 2.66
Sprint 30 m (s)	5.47 (0.55)	4.82 (1.04)	*ns*
Countermovement jump (cm)	32.91 (4.53)	31.12 (4.24)	*ns*
Squat jump (cm)	24.87 (4.23)	25.64 (4.54)	*ns*
Agility T-half (s)	8.05 (1.07)	7.33 (0.71) *	*t* (27) = 2.10
Medicine ball throw 3 kg (m)	8.40 (0.61)	7.69 (0.77) **	*t* (27) = 2.78
Yo-Yo Intermittent Recovery (s)	1040.00 (178.74)	723.08 (220.58) ***	*t* (27) = 4.28

* *p* < 0.05, ** *p* < 0.01, *** *p* < 0.001, ns: non-significant (*p* > 0.05).

**Table 4 ijerph-17-08428-t004:** Summary of the hierarchical regression analyses for anthropometric variables predicting performance parameters. Body height (BH) and body mass (BM) were entered as predictors in step 1, while percent body fat (%BF) was added in step 2. Standardized β coefficient are given for each predictor.

Tests		R^2^	R^2^ Change	F	BH	BM	%BF
Sprint 15 m	Step 1	0.04		0.58	0.15	−0.20	
	Step 2	0.25	0.21	2.81	0.00	−0.04	−0.49 *
Sprint 30 m	Step 1	0.04		0.50	−0.15	0.18	
	Step 2	0.06	0.03	0.58	−0.20	0.24	−0.18
CMJ	Step 1	0.51		13.39	0.47	−0.71 ***	
	Step 2	0.51	0.00	8.58 ***	0.47 **	−0.71 ***	0.01
SJ	Step 1	0.36		7.20 **	0.61 **	−0.36 *	
	Step 2	0.41	0.05	5.70 **	0.68 ***	−0.43 *	0.24
Agility T-half	Step 1	0.41		9.10 **	0.54 **	−0.57 **	
	Step 2	0.51	0.09	8.55 ***	0.43 *	−0.47 **	−0.33 *
Medicine ball throw (3 kg)	Step 1	0.04		0.50	0.16	0.07	
	Step 2	0.40	0.37	5.64 **	-0.05	0.27	−0.65 **
Yo-Yo IR1	Step 1	0.64		22.86 ***	0.40 **	−0.83 ***	
	Step 2	0.79	0.15	30.77 ***	0.27 *	−0.70 ***	−0.41 ***

CMJ: countermovement jump, SJ: squat jump, Yo-Yo IR1: Yo-Yo intermittent recovery level 1, BH: body height, BM: body mass, %BF: percent body fat. Degrees of freedom for the ANOVA F are F (2, 28) at stage 1 and F (3,28) at stage 2. * *p* < 0.05, ** *p* < 0.01, *** *p* < 0.001.

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
