# Peer review of "Body Fat Is a Predictor of Physical Fitness in Obese Adolescent Handball Athletes"

_ijerph, 2020, doi:10.3390/ijerph17228428_

Round 1

Reviewer 1 Report

The objective of the study is very interesting and its aplicability for handball; however, the authors come to very simple conclusions that apply to any sport.

To make observations more specific to handbal, I suggest correlating the bicipital and tricipital folds with throwing ability. Also, it would be interesting to make correlations or associations with the calf crease.

On the other hand, please explain the possible iinfluence of the time of practicing the sport (months or years), between both having both types of players. This, due to the fact that the position in the game, can influence the amount of total body fat mass and that observed in each muscle that is exercised.

Finally, with the observations previosuly described, I suggest writting more sport-specific conclusions. 

Author Response

Reviewer #1

The objective of the study is very interesting and its applicability for handball; however, the authors come to very simple conclusions that apply to any sport.

Author Response (AR)

Thanks a lot for this positive feedback and comment. We changed and improved the conclusions as suggested.

Reviewer #1

To make observations more specific to handball, I suggest correlating the bicipital and tricipital folds with throwing ability. Also, it would be interesting to make correlations or associations with the calf crease.

Author Response (AR)

Thank you for the interesting comment. Indeed, before running the hierarchical regressions, we had initially planned to include all SKF measures in the analysis. We were constrained by the multicolinearity found between %BF and most of the SKF measures. Therefore, we decided to run separately (for exploration purposes) linear regressions to examine the relationship between skinfold (SKF) measures (IV) and each performance variables (DV). The results are presented in the Table below. Combined SKF measures were significantly related to Agility, Medicine Ball throw and Yo-Yo test performances. Specifically, only bicipital and subscapular SKF measures correlated significantly with Agility, while no other specific correlations were detected (only overall). Therefore, bicipital and tricipital SKF were not directly correlated with the throwing performance, which addresses the reviewer’s suggestion. As a result, we decided to exclude this additional analysis from the manuscript that does not add much significance to the interpretations at this stage, especially when considering the hypothesis and main question of the study. We consider that %BF is a measure that combines all SKF variables and represent their sum, independently of specific distribution of fat that, to our knowledge, was not previously addressed in the literature, and therefore hard to speculate/hypothesize about. However, as suggested, we believe that a more focused study on SKF measures (with other important sites) and their relation to performance can be the goal of a future study. We added this in the perspectives of the present study.

Unfortunately, we did not measure other SKF measures during the experiment time and do not have the calf crease measure. The aim of the SKF were mostly to assess %BF, with no specific aim to correlate those with performance measures. A future study can look more specifically into the relationship between performance variables and fat distribution.

Reviewer #1

On the other hand, please explain the possible iinfluence of the time of practicing the sport (months or years), between both having both types of players. This, due to the fact that the position in the game, can influence the amount of total body fat mass and that observed in each muscle that is exercised.

Author Response (AR)

Thanks a lot for this hint. As suggested, we added the mentioned information’s as follows:

Line 307-313: ‘’The possible differences in playing experience (e.g., range: 3-6 years) might have influenced body composition, since players tended to have a higher %BF than the other players. Furthermore, handball-playing position can influence the amount of total %BF as many authors described, that players are significant different regarding body mass depending on the playing position body, weight, BMI, arm span and palm length. Fieseler al. [2] reported that backs, pivots and goalkeepers players are remarkably present greater body mass segments than wings are the shortest players, with the least body weight’’.

Reviewer #1

Finally, with the observations previosuly described, I suggest writting more sport-specific conclusions.

Author Response (AR)

Thanks a lot for this positive feedback and comment. We changed and improved the conclusions as suggested.

Reviewer 2 Report

In my opinion, this is not a novel study. The methodology is correct. The introduction and discussion are very weak. Conclusions should also be improved.

Introduction

Line 34-36 The authors write "... and body mass index (BMI) similar to those of team sport players", and in the bibliography there are 3 articles about handball. What team sports do the authors mean? Please detail the information.

Line 36-37 "Recent studies… [1-5]" ?. Article 5 is from 2002.

"... hypothesis was that body fat would be negatively associated with performance in sprinting, jumping, throwing and running tests, ..." - on what basis was the hypothesis made? In the introduction, there is not a word about the specific dependencies of body fat and sprinting and jumping and throwing and running.

The article uses regression analysis, and in the introduction there is no mention of similar research.

The introduction is intended to show the reader the problem. Describe what has been described in the articles so far. Then identify what this specific study brings to the current state of knowledge. And the introduction proposed by the authors is poor in information and in my opinion does not justify a hypothesis.

Discussion

Line 226-228 This sentence needs to be cited.

"The present study was able to differentiate the unique contribution of% BF to performance parameters in young handball players." - what is this uniqueness? This sentence requires an explanation. And if there is something unique in the research, this thread should be expanded. The development should be based on literature.

"Optimizing body fat should be considered as a training and nutrition goal in order to improve not only the running performance (aerobic capacity and agility), but anaerobic components such as throwing, sprinting and jumping." - move this sentence to the Conclusions section. And if it is to be used in a Discussion, it needs to be cited.

"The latter suggests that previous reports showing significant correlations between BMI and physical fitness do not capture the influence of body fatness separately, and interpretations relative to overweight and obesity based on BMI classifications would merit caution." - it seems to have been researched a long time ago.

“From the literature, it is known that overweight and obese adolescents generally have a poorer performance in motor tests than their normal weight counterparts [34]. A lot of previous research focused on adolescents proved that overweight and obesity negatively affect the level of fundamental motor skills and motor performance [35-36]. Specifically, severe deficits in fundamental movement skills diagnosed as developmental coordination disorder were significantly linked to high percentage of body fat [37, 38]. " - this is an introductory paragraph, so it should be in the Introduction, not in the Discussion. Especially that this paragraph does not contain any information and comparisons to the research contained in this study.

"Findings underline the importance of implementing programs targeting weight / fat control for enhancing and optimizing sports performance. Therefore, handball clubs and other sports institutions should have the onus of developing exercise-training and conditioning programs that are specifically aimed at controlling body mass and fat. " - this sentence is the Practical Application

“Adolescent handball players from the Qatar handball first league presented a% BF of 27.66 (± 293 8.67), which is considered relatively high for young athletic populations. In the below, an attempt to elucidate the effects of excessive fatness on performance parameter among the young handball players is suggested to devise and implement ad-hoc exercise interventions aimed at targeting and managing body mass status. " - this sentence is a Conclusion, an Application, not a Discussion.

In my opinion, the discussion should be deeply rewritten.

Conclusions

“Adolescent handball players with an obesity classification based on% BF showed lower performance in fundamental physical fitness tests. Specifically, the increase of% BF seemed to have deleterious effects on throwing and running performance after controlling for body height and mass. " - these are the Results and not the Conclusions

“We assume that not only body mass status, but also the ratio between the amount of body fat and lean mass plays a crucial role in developing athletes. - on what basis this conclusion? It results from these studies?

Author Response

Reviewer #2

In my opinion, this is not a novel study. The methodology is correct. The introduction and discussion are very weak. Conclusions should also be improved.

Author Response (AR)

Thank you for your remark. In response to your queries theses sections are now largely modified. In addition, the conclusions was reworked and improved.

Introduction

Reviewer #2

Line 34-36: The authors write "... and body mass index (BMI) similar to those of team sport players", and in the bibliography there are 3 articles about handball. What team sports do the authors mean? Please detail the information.

Author Response (AR)

Thanks a lot for this valuable hint. We have detailed the information as follows:

Line 41: ‘’….similar to those of team sport players (e.g., Rugby and American football) [3-7]’’.

Reviewer #2

Line 36-37 "Recent studies… [1-5]" ?. Article 5 is from 2002.

Author Response (AR)

Thanks a lot for this valuable comment. We have used a new recent reference from 2017 as suggested:

‘’Schwesig, R.; Hermassi, S.; Fieseler, G.; Irlenbusch, L.; Noack, F.; Delank, K.S.; Shephard, R.J.; Chelly. M.S. Anthropometric and physical performance characteristics of professional handball players: influence of playing position. J Sports Med Phys Fitness 2017, 57,1471-1478’’.

Reviewer #2

"... hypothesis was that body fat would be negatively associated with performance in sprinting, jumping, throwing and running tests, ..." - on what basis was the hypothesis made? In the introduction, there is not a word about the specific dependencies of body fat and sprinting and jumping and throwing and running.

Author Response (AR)

Thank you for your relevant remark. In response to your comment, the introduction was rewritten, to better support the hypothesis of the study. We have also clarified and introduced the topic of obesity in adolescent athletes. Therefore, the following paragraph was added in the text :

Line 83-90: ‘’It seems that an increased %BF presents different implications for running and jumping performances among different age groups in athletes and this must be taken into account by coaches and trainers. Furthermore, there is a need to clarify the associations between factors such as %BF and fitness affecting the physical performance in team handball players. Since, the overall overweight and obesity prevalence among male adolescents in Qatar was higher [28] than the global prevalence of obesity and overweight (18%) reported by the WHO in 2016. Due to overcome sedentary lifestyles, the number of overweight adolescent in Qatar is increasing every day while their level of physical activity and motor skills declines’’.

Reviewer #2

The article uses regression analysis, and in the introduction there is no mention of similar research.

Author Response (AR)

While several studies that we mention have looked into the relationship between anthropometric variables and fitness / performance measures using simple Pearson correlations [6,8], Linamost of the studies presented in the introduction use regression analyses. For example, Zapartidis et al. (2011) [10] have used multiple regression analysis to examine determinants of ball velocity from anthropometric variables amongst others. Morever, Huang and Malina (2010) [14] used regressions to model the relationship between BMI and physical fitness in Taiwanese youth. Specifically, one of the recent studies (Hammami et al., 2019) [11] have used multiple linear regressions with a similar hierarchical model. Authors tested the relationship between maturity and anthropometric variables (being predictors) and physical performance (being DV). They did so in steps in order to look for the unique contribution of anthropometric measures after accounting for maturity (age). We believe that our choice of a statistical approach is justified in the methods and an efficient way to answer our hypothesis.

Reviewer #2

The introduction is intended to show the reader the problem. Describe what has been described in the articles so far. Then identify what this specific study brings to the current state of knowledge. And the introduction proposed by the authors is poor in information and in my opinion does not justify a hypothesis.

Author Response (AR)

Thank you for your remark. In response to your queries a new section was created related to introduction and the following paragraph was added in the text.

Line 71-82: ‘’Furthermore, it is needed to clarify the associations between anthropometric characteristics (i.e., body height and mass and %BF) and physical fitness in adolescent handball players, reflecting performance parameters.

Consequently, the results in this direction seem the potential benefits of long-term athletic development as a pathway that could enhance the health, fitness, and performance of adolescent players [26,27]. However, the development of sporting talent is very important, highly valued, and extremely rewarding for both athletes and practitioners alike; thus, however, it is imperative from a public health perspective that a structured, progressive, and integrated approach to youth training is viewed as a developmental pathway for players and adolescents of all ages and abilities. Likewise, this research area scarcely concerns countries from the Gulf region, and this geographical area has unique features, which means research from other parts of the world may not be applicable to the Gulf’’.

Discussion

Reviewer #2

Line 226-228 This sentence needs to be cited.

Author Response (AR)

As suggested, we have added the following references as suggested:

  • ‘’Mondal H, Mishra SP. Effect of BMI, Body Fat Percentage and Fat Free Mass on Maximal Oxygen Consumption in Healthy Young Adults. J Clin Diagn Res. 2017;11(6):CC17-CC20’’.
  • ‘’The Relationship between Body Fat Percentage and Some Anthropometric and Physical Fitness Characteristics in Pre- and Peripubertal Boys.’’

Reviewer #2

"The present study was able to differentiate the unique contribution of% BF to performance parameters in young handball players." - what is this uniqueness? This sentence requires an explanation. And if there is something unique in the research, this thread should be expanded. The development should be based on literature.

Author Response (AR)

Unique contribution” is a statistical term used to describe the unique proportion of variance explained by a specific factor (as opposed to the relative contribution), when accounting for other factors. As such, the measure of unique contribution (R2 change) at each step of the model represents the relationship between the added predictor (%BF – step 2) and the part of the outcome that is not explained by the other predictors in the model (step 1). The literature indicates that performance measures are affected by BMI and therefore body mass and height. From a statistical point of view, those variables that are known to affect performance should be included in the model first, before adding %BF that is being explored here.

Reviewer #2

"Optimizing body fat should be considered as a training and nutrition goal in order to improve not only the running performance (aerobic capacity and agility), but anaerobic components such as throwing, sprinting and jumping." - move this sentence to the Conclusions section. And if it is to be used in a Discussion, it needs to be cited.

Author Response (AR)

Thank you for your relevant suggestion. The sentence was moved to the Conclusions section as suggested.

Reviewer #2

"The latter suggests that previous reports showing significant correlations between BMI and physical fitness do not capture the influence of body fatness separately, and interpretations relative to overweight and obesity based on BMI classifications would merit caution." - it seems to have been researched a long time ago.

Author Response (AR)

As suggested by the reviewer, the statement should be re-written to address specifically gaps in the literature that are mentioned in the introduction. Although several studies have looked into the relationship between weight-status (BMI) or even %BF and physical fitness in adolescents, none to our knowledge have studied athletic adolescent populations or even isolated the contribution of fatness separately from BMI or BM/BH. Therefore, the sentence was re-written to be more specific:

Line 319-322 : “The latter suggests that previous reports showing significant correlations between BMI and physical fitness do not capture the influence of body fatness separately, and interpretations relative to overweight and obesity based on BMI classifications would merit caution in athletic populations“.

Reviewer #2

“From the literature, it is known that overweight and obese adolescents generally have a poorer performance in motor tests than their normal weight counterparts [34]. A lot of previous research focused on adolescents proved that overweight and obesity negatively affect the level of fundamental motor skills and motor performance [35-36]. Specifically, severe deficits in fundamental movement skills diagnosed as developmental coordination disorder were significantly linked to high percentage of body fat [37, 38]. " - this is an introductory paragraph, so it should be in the Introduction, not in the Discussion. Especially that this paragraph does not contain any information and comparisons to the research contained in this study.

Author Response (AR)

Thank you for your relevant suggestion. The paragraph was moved from the discussion to the introduction section.

Reviewer #2

"Findings underline the importance of implementing programs targeting weight / fat control for enhancing and optimizing sports performance. Therefore, handball clubs and other sports institutions should have the onus of developing exercise-training and conditioning programs that are specifically aimed at controlling body mass and fat. " - this sentence is the Practical Application.

Author Response (AR)

As suggested, the sentence was moved to the new ‘’Practical Application’’ section.

Reviewer #2

“Adolescent handball players from the Qatar handball first league presented a% BF of 27.66 (± 293 8.67), which is considered relatively high for young athletic populations. In the below, an attempt to elucidate the effects of excessive fatness on performance parameter among the young handball players is suggested to devise and implement ad-hoc exercise interventions aimed at targeting and managing body mass status. " - this sentence is a Conclusion, an Application, not a Discussion.

Author Response (AR)

We agree with the reviewer. The sentence was used in the ‘’Practical Application’’ instead.

Reviewer #2

In my opinion, the discussion should be deeply rewritten.

Author Response (AR)

We have reworked on the discussion as proposed. We believe that is it improved in the revised manuscript.

Reviewer #2

“Adolescent handball players with an obesity classification based on% BF showed lower performance in fundamental physical fitness tests. Specifically, the increase of% BF seemed to have deleterious effects on throwing and running performance after controlling for body height and mass. " - these are the Results and not the Conclusions.

Author Response (AR)

Results point to a specific relationship between anthropometric measures and each of the fitness test performed separately. The interpretation considers those tests as being clusters, and groups them into running, throwing and jumping performances for example. We believe that the statement is important for the conclusion as it reports main findings that are worth being repeated in the conclusions; especially for readers who will directly jump to that section.

Reviewer #2

“We assume that not only body mass status, but also the ratio between the amount of body fat and lean mass plays a crucial role in developing athletes. - on what basis this conclusion? It results from these studies?

Author Response (AR)

This conclusion is based on the fact that we find %BF to have a unique contribution to performance that is not related to the contributions of BM and BH. Therefore, we only assume using the logic of a two component model of body composition (BM = BF + FFM) that not only %BF but also %FFM (lean mass) would affect performance. This is only an assumption and not directly a reported result, in an effort to push the interpretations. 

Reviewer 3 Report

This is an interesting study and the results are worthwhile for clinicians, trainers and coaches involved in handball. However, the authors should clarify their definition of obesity. Furthermore, the reason why the authors used % body fat for the classification should also be clearly explained. % body fat and BMI of non-obese group should be categorized as overweight.

Line 50: Please be more specific about the fitness tests to better enable readers to understand the association more clearly.

Line 56: How popular is handball? Perhaps you could include some information in the 1st paragraph.

Line 70: BMI of non-obese group was 26.12, which should also be classified as overweight. What is the definition of obesity in this study? On what basis were the participants divided into 2 groups (obese and non-obese)? Obesity should be divided depending on BMI not body fat.

Line 94: Even in non-Obese subjects, when the % body fat is 21.66 (4.70), the subject should be classified as overweight.

Line 107: M2 should be changed to m2.

Line 108: The reliability of the measurements using Harpenden calipers should be determined. How many examiners were required to compile the measurements?

Line 135: “10 min” should be “10-min.”

Line 148: “5 min” should be “5-min.”

Line 197: “ps>0.05” should be “p>0.05.”

Line 201: “15 m” should be “15-m.”

Line 203: In Table 3, “n=16” of non-obese should be 1 line below. The number of significant digits with squat jump should be fixed.

Line 209: “15 m and 30 m” should be “15-m and 30-m.”

Line 238-239: This classification should be explained in the Method section.

Line 300: Are the authors able to provide the peak height velocity (PHV) about their maturity status of subjects?

Author Response

Reviewer #3

This is an interesting study and the results are worthwhile for clinicians, trainers and coaches involved in handball. However, the authors should clarify their definition of obesity. Furthermore, the reason why the authors used % body fat for the classification should also be clearly explained. % body fat and BMI of non-obese group should be categorized as overweight.

Author Response (AR)

The cut-off values considered for adolescents’ obesity are now presented in the Methods. Given that 29 participants volunteered in the study, classifications were made in two groups (i.e., obese vs non-obese) in order not to further decrease the sample size in each group. Our groups were specifically named “Obese” and “non-obese” as opposed to “obese” vs “normal-weight” or “healthy weight”. We have considered all participants that do not meet the criterion for obesity as “non-obese”.

Given that participants in the present study were adolescents, aged 15-16y and 18-19y, two different cut-off values were used (as those differ by age). For those aged 15 and 16 years, obesity was considered for %BF values above 31%. For those aged 18 and 19 years, obesity was considered for %BF values of 22% and above. Those classifications were considered based on Heyward & Gibson, 2018 (7th edition, Advanced Fitness Assessment and Exercise Prescription, p. 220, Human Kinetics). Accordingly, obesity is clearly defined for ≤17y and 18-34y. Below the obesity classification, high, mid and low %BF classifications are identified. In our sample, only one participant of the 16 in the non-obese group had a %BF marked as “high” (i.e., overweight), all other subjects were in the mid-ranges (i.e., normal %BF, non-overweight).

Cut-off values according to age groups were added to the manuscript and the reference was updated.

In Qatar, there are no studies that present norms of children or adolescents’ %BF to facilitate referencing. Only recently, a new study in Qatar (Bawadi et al., 2020, Age and gender specific cut-off points for body fat parameters among adults in Qatar, Nutrition Journal) revealed a cut-off value of 35.1% for obesity classification in adults below the age of 40 years. %BF values in Qatar seem to be higher than global reported values.

We did not use BMI classifications (and decided to collect 4 skinfold measures from participants) because we believe that BMI values can be biased for athletes that are expected to have higher fat-free mass (muscles), and therefore body mass.

Reviewer #3

Line 50: Please be more specific about the fitness tests to better enable readers to understand the association more clearly.

Author Response (AR)

According to the reviewer’s suggestion we added the following in the text:

Line 55-56: ‘’Generally, a negative correlation between %BF and performance on several fitness tests (e.g., Fist ball throw, standing long jump, 30 m Dash and 1200 m Running) was indicated in 9-13 year-old boys [15]’’.

Reviewer #3

Line 56: How popular is handball? Perhaps you could include some information in the 1st paragraph.

Author Response (AR)

According to the reviewer’s suggestion we added the following text in the 1st paragraph:

Line 36-37: ‘‘Handball is an Olympic sport played worldwide and at a highly professional level in many countries‘‘.

In addition we have added the following in the text:

Line 67: ‘‘Considering the popularity of handball as the number two sport in the world...‘‘.

Reviewer #3

Line 70: BMI of non-obese group was 26.12, which should also be classified as overweight. What is the definition of obesity in this study? On what basis were the participants divided into 2 groups (obese and non-obese)? Obesity should be divided depending on BMI not body fat.

Author Response (AR)

Kindly refer to our answer on the first comment.

Reviewer #3

Line 94: Even in non-Obese subjects, when the % body fat is 21.66 (4.70), the subject should be classified as overweight.

Author Response (AR)

Please refer to our answer on the first comment.

Reviewer #3

Line 107: M2 should be changed to m2.

Author Response (AR)

Corrected as suggested

Reviewer #3

Line 108: The reliability of the measurements using Harpenden calipers should be determined. How many examiners were required to compile the measurements?

Author Response (AR)

Thank you for your comment. The methods section stating the measurements of skinfolds was rewritten, underlying the procedures used to avoid the errors of measurement. Furthermore, we have provided more information about the details of the equation used and reported in adolescent athletes for measuring the percentage of body fat. Following your suggestions, we changed and clarify our procedure as follows:

Line 148-156:‘’Skinfold thickness was measured to the nearest 0.1 mm with a Harpenden caliper. Duplicate readings were taken at each site, and the average of the two was recorded. If the two readings differed by more than 2 mm, a third, one was taken, and the closest two were averaged. The sum of the four skinfold measurements was used as an estimate of body fat according to the sex- and age- specific Durnin-Womersley equation [19] as previously reported in adolescent handball players (Chelly al., 2014). However, the overall percentage of body fat was estimated from the biceps, triceps, subscapular, and suprailiac skinfolds, using the equation [19]’’:

% Body fat = (4.95/ (Density - 4.5)) ● 100

Where Density = 1.162–0.063 (LOG sum of 4 skinfolds).

Reviewer #3

Line 135: “10 min” should be “10-min.”

Author Response (AR)

Corrected as suggested

Reviewer #3

Line 148: “5 min” should be “5-min.”

Author Response (AR)

Corrected as suggested

Reviewer #3

Line 197: “ps>0.05” should be “p>0.05.”

Author Response (AR)

ps is the plural form of p (p-values) and conventionally used when many p values are presented.

Reviewer #3

Line 201: “15 m” should be “15-m.”

Author Response (AR)

Corrected as suggested

Reviewer #3

Line 203: In Table 3, “n=16” of non-obese should be 1 line below. The number of significant digits with squat jump should be fixed.

Author Response (AR)

Corrected as suggested

Reviewer #3

Line 209: “15 m and 30 m” should be “15-m and 30-m.”

Author Response (AR)

Corrected as suggested

Reviewer #3

Line 238-239: This classification should be explained in the Method section.

Author Response (AR)

The classification is now presented in the methods section

Reviewer #3

Line 300: Are the authors able to provide the peak height velocity (PHV) about their maturity

Author Response (AR)

Unfortunately, data related to maturity were not collected at the moment of the experiment, and PHV is not available to authors.

Round 2

Reviewer 1 Report

The authors adequately presented information that contextualizes the health status of the players, with respect to the world population. They, also, improved the description of anthropometric measurements. Finally, the conclusions are clearer and highlight the application of the results.

Reviewer 2 Report

I accept the article in a current form.

Reviewer 3 Report

None